# Comparison of Different Optimization Techniques for Model-Based Design of a Buck Zero Voltage Switching Quasi-Resonant Direct Current to Direct Current Converter

**Nikolay Hinov** [1,*] and **Bogdan Gilev** [2]

1    Department of Power Electronics, Technical University of Sofia, 1000 Sofia, Bulgaria
2    Department of Mathematical Modeling and Numerical Methods, Technical University of Sofia, 1000 Sofia, Bulgaria; b_gilev@tu-sofia.bg
*    Correspondence: hinov@tu-sofia.bg; Tel.: +359-296-52569

**Abstract:** The present paper provides a comparison of different optimization techniques applied to the model-based design of a Buck Zero Voltage Switching (ZVS) Quasi-Resonant DC-DC Converter. The comparison was made both on the basis of the duration of the optimization procedures and in terms of guaranteeing the performance of the power electronic device. The main task of the paper is to present various techniques based on the use of mathematical software for the optimal design of Quasi-Resonant DC-DC converters. These topologies were chosen because in them, the design is carried out according to computational procedures, in which several iterations are often necessary for the successful completion of the process. An optimization procedure with a target function reference curve of the output voltage was used. In this way, the optimization is performed without the need for a complete design of the device but only by using base ratios, design constraints, and past experience to determine initial values and intervals of change in circuit parameters. This is also the main advantage of the used optimization of the reference curve type of the output, compared to applying other objective functions, such as achieving minimum losses or maximum efficiency of the device.

**Keywords:** Buck Quasi-Resonant DC-DC converter; Matlab; model-based design; optimization; Zero Voltage Switching (ZVS)

## 1. Introduction

Quasi-Resonant direct current to direct current (DC-DC) converters are a special type of DC-DC converters that use the principle of resonance to achieve higher efficiency and lower losses when converting electrical energy from one direct current to another. The main idea behind applying Quasi-Resonant DC-DC converters is to combine the characteristics of resonant and non-resonant converters [1,2]. They typically operate at a switching frequency close to the resonant frequency, with the ratio between these two frequencies being optimized by design for a particular application. This type of power electronic device allows for operation with greater energy conversion efficiency and less loss compared to classic DC-DC converters.

Quasi-Resonant DC-DC converters are used in a variety of applications, including power electronics, industrial systems, medical devices, and more. It can be implemented in different configurations such as converter topology (e.g., Boost, Buck, and LLC converters) and operate accordingly with different settings and currents depending on the specific application requirements [3].

On the other hand, Quasi-Resonant DC-DC converters have their advantages and disadvantages, which must be taken into account when choosing to use this type of

converter for a specific task. The advantages of Quasi-Resonant DC-DC converters are as follows [1,3]:

- Higher efficiency: Quasi-Resonant converters can achieve higher efficiency compared to traditional DC-DC converters. This means less energy loss during its conversion.
- Lower Electromagnetic Interference (EMI): They are designed to reduce EMI, making them suitable for applications where it is important to comply with high requirements of electromagnetic standards and regulations.
- Suitable for high powers: Quasi-Resonant converters can be used to convert high powers and high voltages.
- Smaller weight and dimensions: they have smaller dimensions and lighter components compared to some other types of DC-DC converters.

  Disadvantages of Quasi-Resonant DC-DC converters are related to [3] and are as follows:

- Complex control system: The design and control of quasi-resonant converters require more complex electronics to monitor the zero crossing of the resonant current and/or voltage. This complicates the synthesis of control by the designer, which can increase the costs of design, setup, and operation and, accordingly, the complexity of the controller and the device as a whole.
- Higher costs: due to the complexity of the design and control, Quasi-Resonant converters are usually more expensive compared to classical DC-DC converters.
- More limited operating frequencies and corresponding output voltage adjustment range: due to the on-time or off-time requirements of the semiconductor switch, they generally operate at a more limited frequency range than other types of converters.
- Higher part-load losses: Quasi-Resonant converters can have higher part-load losses, which must be taken into account when designing the device.

The choice of a Quasi-Resonant DC-DC converter for a power electronic device implementation is highly dependent on the specific requirements and constraints for a given application, and in this respect, careful analysis and evaluation must be carried out to determine whether this type of converter is best suited for the particular task.

On the other hand, the analysis of Quasi-Resonant DC-DC converters, like most power electronic devices, is carried out on the basis of determining the main relationships between the state variables in an established mode of operation when the process of energy accumulation in the inductances and capacitors in the circuit has been completed. In this way, approximate design methodologies are obtained that do not reflect the dynamics and different possible operating modes. In this sense, devices designed on the basis of established processes do not have adequate behavior in transient modes: during start-up, when changing the input voltage, or the load current.

### 1.1. Literature Review

Due to the growing need to apply Quasi-Resonant DC-DC converters, a number of authors present results based on the study of established modes of operation with or without the use of a controller. In [4], the step-by-step analysis, the design methodology, and the results of experiments on an experimental sample of the Boost Quasi-Resonant DC-DC converter are given in detail. By using a specialized integrated circuit in the control synthesis, very good stability of the output voltage has been achieved against various disturbing effects. In addition, the authors have achieved a relatively high level of efficiency at full load of the power circuit—93.5%. Similar studies, but on a Buck Quasi-Resonant DC-DC converter, are given in [5]. What is specific in this particular case is the achievement when turning on the transistor to work simultaneously with ZVS and ZCS, and the switching loss is eliminated. On the basis of analytical ratios found, the connection between the parameters of the resonant circuit and the switching-on time of the transistor was found to ensure operation with soft commutations. According to the presented expressions, a 3 kW sample was designed and prototyped with a maximum efficiency of 98.7%. In ref. [6], the analysis and design of a two-half-period Quasi-Resonant Buck ZVS

DC-DC converter is presented. Via the obtained analytical expressions, the converter is designed and prototyped. The test results confirm the adequacy of the analysis and the corresponding design procedure. Another main problem, due to the specifics of control, is the synthesis and adjustment of the controller of the Quasi-Resonant DC-DC converters. In [7], a comparison between two types of Buck Quasi-Resonant DC-DC converter control is presented: linear feedback control, based on the tracking of the average value of the current through the filter inductance, and current-mode control. In this aspect, in addition to analytical dependencies, the results of prototype testing are presented, through which the theoretical conclusions and inferences are validated. A further increase in efficiency is achieved by using synchronous topologies, as shown in [8]. A new synchronous Buck ZVS DC-DC converter is proposed there, with soft commutations realized over the entire range of load changes. The suggested Buck Quasi-Resonant DC-DC converter consists of a Buck conventional synchronous DC-DC converter with additionally added resonant elements. The operability and efficiency of this circuit have been validated via experiments on a laboratory sample. Achieving the set quality parameters both in terms of the power part and in terms of the controller is connected with the modeling of the Quasi-Resonant DC-DC converters. Models of the three most commonly used transformerless DC-DC converters are presented in [9] as follows: Buck, Boost, and Buck–Boost. Via derived analytical expressions based on state-space averaging, the transmission functions were determined, and a control synthesis was made using the voltage–age–mode control method. With the implemented models, the stability of the open and closed systems was investigated, in which the DC-DC converter was modeled. Another modeling approach based on a description of the electromagnetic processes in a Quasi-Resonant DC-DC converter is given in [10]. The differential equations concerning the state variables, together with logic equations that set the switching conditions for ZVS mode operation, are implemented in Matlab/Simulink. Thus, the created model describes the operation of the device in both quasi-steady and dynamic modes, and the use of Simulink allows the easy implementation of the model by non-specialists in the field of computational mathematics, which is convenient for use by designers and students. The analysis of the presented sources gives reason to summarize that due to the peculiarities of the operating modes, the use of innovative methods is necessary for the successful design and prototyping of Quasi-Resonant converters since the classic ones are not able to fully guarantee the requirements and quality indicators for modern power electronic devices. One of the possible new approaches in the study of this class of schemes is the application of model-based optimization. In this sense, the use of optimization methods in the design of power electronic devices and systems is one of the most promising directions in power electronics. A detailed review of various optimization techniques applied in the development of power electronic devices with different purposes is given in [11]. There is proposed a classification of optimization techniques from the point of view of their application in power electronics. Examples of optimal design of devices with diverse applications and purposes are presented and summarized. Ref. [12] proposed an optimal design approach using the preliminary design of power electronics. Generally, the optimization problem is solved on a virtual prototype, and to ease the computational procedures, all state variables are assumed to be continuous. The proposed method is illustrated by the virtual design of an Interleaved Buck DC-DC Converter. The authors have also emphasized the possible applications of the presented methodology for rapid design and prototyping, such as the comparison of several operating modes; validation of various candidate technologies for prototyping; and negotiation of converter specifications with project sponsors. In [13], an approach to optimize the design of a power electronic device using a continuous variable is presented. An objective function was used to minimize the total cost of the components that make up the device under consideration. The advantages of this methodology are demonstrated with an example of the design of a Boost DC-DC converter for power factor correction and reduced levels of electromagnetic interference with the addition of an input filter on the supply network side. Using optimization techniques, the values of the building elements have been determined, with which the total

cost of these components is minimal, taking into account the structural limitations and complying with the requirements of the design task. Better results for optimal design have been achieved by applying multi-objective optimization [14]. This paper presents a multi-objective optimization method by modeling power electronic converters using polynomial functions. Thus, the multi-objective optimization of converters can be formulated as a geometric program in the form of a convex optimization problem. This makes it possible, via the use of fast and powerful computing tools, to guarantee the optimality of solutions in a global sense, i.e., to have a guarantee of obtaining a decision that is not of a local nature. The method is illustrated by the optimal design of low-power multistage flying capacitor step-down converters. The presented results prove that using geometric programming, sets of circuit parameter values are obtained that are globally Pareto-optimal in a very short time—as little as 25 s when working with a mid-range to the high-end notebook computer. In this way, optimal designs for three different topologies of force resistance for multiple design spaces can be determined in a few hours, which is a significant advantage over other optimization methods currently in use. In ref. [15] considered the optimal design of DC-DC converters for high-power and low-voltage applications such as decentralized electric power generation systems (photovoltaic arrays and fuel cell stacks). One of the main barriers to the development of power electronic converters in renewable energy systems is the achievement of high efficiency. In this work, several optimally designed topologies are compared in terms of efficiency and losses: Isolated Flyback Boost DC-DC Converter, conventional Boost DC-DC converter and Interleaved Boost DC-DC converter. In this case, the optimization also includes choosing the appropriate technology for the semiconductor switches: silicon metal-oxide field-effect transistor (Si-MOSFET) and gallium nitride high electron mobility transistor (GaN-HEMT). An improved full-bridge (ZVS) DC-DC converter is proposed in [16], with a wide range of input voltage and output current variation. Optimal design is based on a target function to achieve maximum efficiency. The methodology was tested using a 1.2-kW/105-kHz prototype, which achieved an efficiency higher than 95% at full load. Works by [17–19] considered the optimal design of DC-DC converters using LLC topology. These power electronic devices are also characterized by a complex analytical description of the electromagnetic processes in the resonant circuit and the variety of operating modes. In this sense, they are characterized by complex design methods, which also include recusal procedures depending on the fulfillment or non-fulfillment of certain conditions valid for securing work in certain regimes. In ref. [17], an original topology designed to control energy flows between energy storage systems and a DC microgrid is presented. A method is proposed to optimize the circuit elements to achieve a reduction in resonant current oscillations in the AC circuit of the device while maintaining soft commutations. Via experiments with a laboratory prototype, the efficiency improvement by reducing the losses in the semiconductor switches has been demonstrated. The optimal design of a bidirectional LLC resonant DC-DC converter for the solid-state transformer (SST) implementation is given in [18]. The selection of the bidirectional LLC converter was made on the basis of achieving higher efficiency with a wider range of zero-voltage switching (ZVS), zero-current switching (ZCS), and a correspondingly simpler control strategy. Higher efficiency is achieved by using a device design optimization procedure, using an objective function to minimize power losses in the circuit elements. In ref. [19], to achieve zero voltage (ZVS) and zero current (ZCS) switching, an optimal design of a half-bridge LLC resonant DC-DC converter is presented. This was achieved by optimizing the parameters of the resonant LLC circuit using a model based on the fundamental harmonic approximation (FHA) method. The experimental results give reason to claim that the proposed optimization procedure is an effective design tool. In refs. [20,21], the authors consider the optimal design of resonant DC-DCs with an application for charging electric vehicles. To achieve this goal, an approach for the most thorough analysis of complex power circuit topologies based on the Fundamental Harmonic Approximation (FHA) method and subsequent optimization procedures to ensure both soft commutation and reference efficiency is used of high efficiency with a wide range of load variation, which is character-

istic of the batteries of electric vehicles. The entry of artificial intelligence into the field of engineering research is also reflected in the optimal design of power electronic devices. In refs. [22–24], a very detailed review of different artificial intelligence techniques applicable to the realization of electronic converters with guaranteed performance under different output conditions and specific application requirements is presented. Various optimization techniques applied in the different phases of the life cycle of power electronic devices and systems, design, prototyping, and operation are discussed in great detail. On the other hand, methods for the optimal design of electronic converters based on machine learning have already been proposed and experimentally confirmed [25,26]. The main problem in applying machine learning is related to collecting sufficiently representative data with which to perform the training. Most often, these data are collected by using mathematical models. Ultimately, the main challenges to the application of these innovative methods are related to the normalization of the data and the selection of an adequate structure to be trained with a suitable data set.

*1.2. Conclusions, Purpose, and Tasks of the Study*

From the analysis of the published results related to the optimal design of power electronic devices, the conclusion follows that, despite the wide variety of topologies and applications, target functions related to achieving maximum efficiency or minimal losses in devices are most often used. The realization of the specific optimization procedures is related to a good mastery of the mathematical apparatus and the means of mathematical modeling.

On the other hand, solving multi-objective optimization involves finding the best solution that satisfies several criteria, often with conflicting interests. Here are some key aspects and methods that are used in solving tasks related to multi-criteria optimization:

- Concept of optimality: In multi-criteria optimization, the concept of optimality is different from that in single-criteria optimization. Usually, a "Pareto optimal" solution is sought where one criterion cannot be improved without worsening another.
- Solving techniques: There are various techniques for solving multi-criteria problems, such as Linear programming: used for problems with linear constraints and objective functions; Compromise programming methods: these methods look for a solution that minimizes the difference (trade-off) between different criteria; and Evolutionary and Genetic Algorithms: suitable for complex tasks where traditional methods are ineffective.
- Pareto set: This is the set of all Pareto optimal solutions. Choosing the "best" solution from this set often depends on the preferences of the decision maker.
- Trade-offs: In multi-criteria optimization, it is often necessary to make trade-offs between different criteria in order to reach a satisfactory solution.
- Software tools: There are various software tools and packages that can help solve multicriteria tasks, such as MATLAB [27], Python libraries e.g., Pyomo [28], and other specialized tools.

It is important to note that the choice of an appropriate method depends on the specifics of the task, including the number of criteria, the type of constraints, and the form of the objective functions.

In this sense, the main task of the presented research is, based on the modern achievements of mathematical modeling, computational mathematics, and mathematical software, to present and compare different approaches for reducing multi-criteria optimization to a single-criteria one based on a reference curve with respect to the output voltage. Via a specific computational example, the transformation methods that give the best results for achieving the optimal design of power electronic devices are evaluated.

The paper is organized as follows: the first part describes the main challenges and problems in realizing the optimal design of Quasi-Resonant DC-DC converters and the various optimization methods. Examples of optimal design of the class of devices under consideration are presented. The second part is devoted to the modeling of the Buck ZVS Quasi-Resonant DC-DC converter. In the third part, a methodology for the design of the Buck ZVS Quasi-Resonant DC-DC converter is presented, and on this basis, the initial values of the circuit elements are determined. In the fourth part, a two-criteria optimization problem with constraints is formulated, which is applied to determine the capacitors and inductances in the power circuit. In the fifth section, the two-criterion optimization is reduced by various methods to a one-criterion with constraints. In the sixth and seventh sections, the discussion, conclusion, and directions for research development are given. All optimization tasks are solved on the basis of algorithms and author's programs, which determine optimal values of the circuit elements in a pre-set interval based on various considerations.

## 2. Modeling of Buck ZVS Quasi-Resonant DC-DC Converter

The schematic of the Buck ZVS Quasi-Resonant DC-DC Converter is shown in Figure 1. It is composed of two main groups of elements: basic ones that form the classic Buck DC-DC converter—transistor, diode, filter inductor, and filter capacitor; and resonant—resonant inductance and resonant capacitor, through which, subject to certain conditions, soft commutations (ZVS) are realized. The following designations are used: $U_d$—input voltage; $L_r$—resonant inductance; $C_r$—resonant capacitance; $L_f$—filter inductance; $C_f$—filter capacity; *Rload*—load resistance; $T = 1/f$—period, respectively, switching frequency of the transistor; and pulse generator—a generator of control pulses with a Duty cycle D.

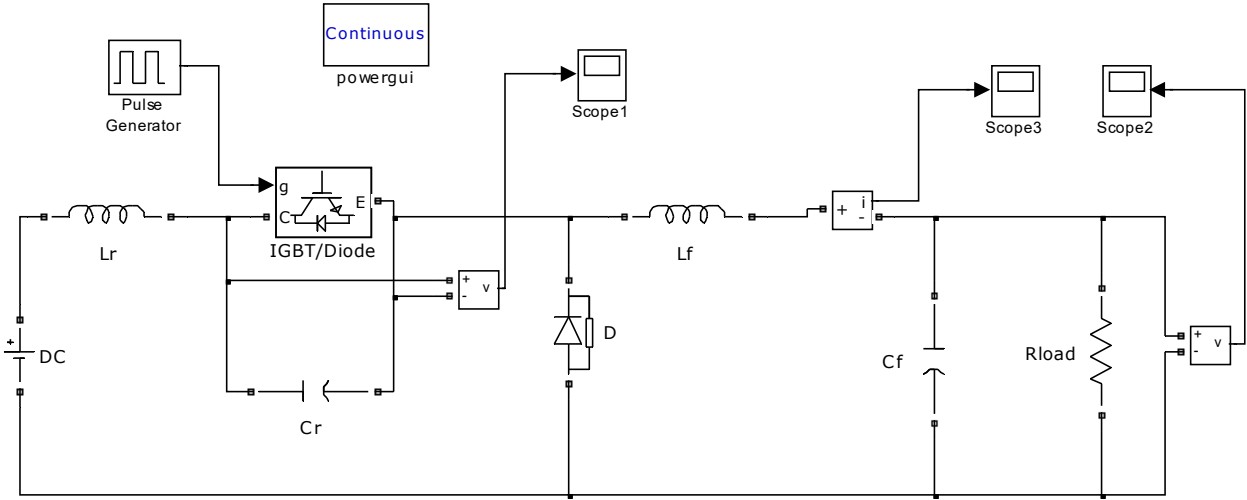

**Figure 1.** Buck ZVS Quasi-Resonant DC-DC Converter.

The operation of the power circuit is discussed in detail in [2], and generally, during operation, there are four intervals: of energy accumulation in the resonant inductance; of conducting the transistor with the diode off; of charging the resonant capacitor; and of the occurrence of a resonance process, in the successive resonance circuit composed of the resonance elements. In this sense, the analysis of the DC-DC converter is carried out in an established mode after the completion of the transient processes in the power circuit, and in this way, different transfer functions are obtained depending on the modes of operation—one half-period or two half-period.

Due to the specific structure and operating modes of the power circuit, the diode must conduct both forward and reverse, and this must be reflected appropriately. For this, the circuit of Figure 1 is modified with an equivalent circuit shown in Figure 2.

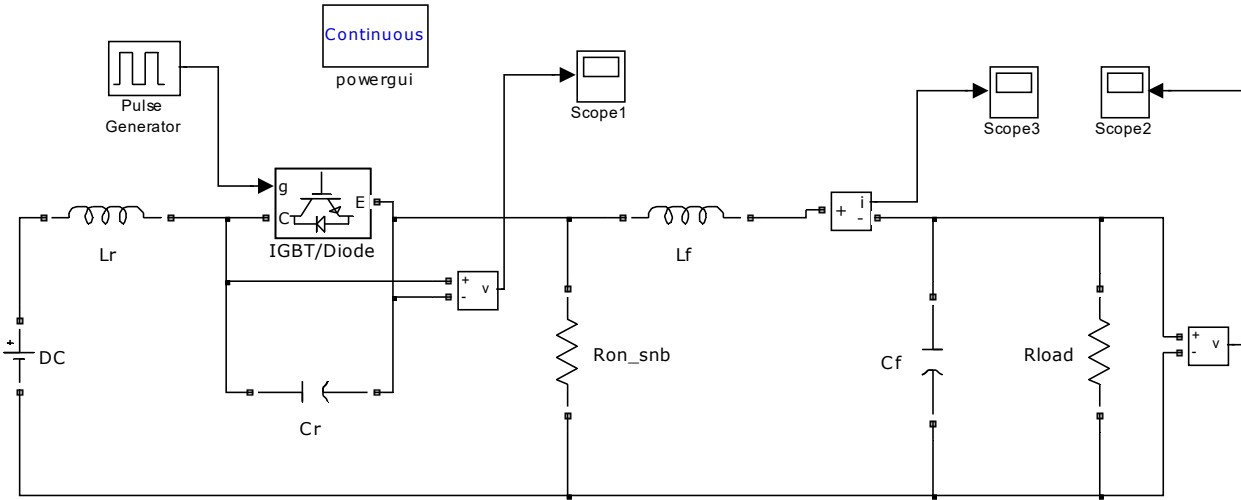

**Figure 2.** Buck ZVS Quasi-Resonant DC-DC converter equivalent circuit.

Diode D is replaced by a resistance $R_{on\_snb}$, which changes its value depending on what is the ratio of the currents that flow through it. Thus, this resistance is defined by the following conditions:

$$R_{on\_snb} = \begin{cases} 500, & for\ i_{Lr} - i_{Lf} \geq 0 \\ 0.01, & for\ i_{Lr} - i_{Lf} < 0 \end{cases}, \tag{1}$$

where $i_{Lr}$ and $i_{Lf}$ are the respective currents through the resonant and filter inductances, and according to Kirchhov's laws, the current through the diode is equal to their difference.

The system from Figure 2 has a variable structure, and for its modeling, the switching functions $K_0$ and $K_1$, and Kirchhov's laws were used. In this way, the following mathematical model describing the operation of the studied power electronic device was obtained:

$$\begin{aligned} L_r \frac{di_r}{dt} + K_0 \cdot u_{Cr} + R_{on\_snb}(i_{Lr} - i_{Lf}) &= U_d \\ L_f \frac{di_{Lf}}{dt} &= -u_{Cf} + R_{on\_snb}(i_{Lr} - i_{Lf}) \\ C_r \frac{du_{Cr}}{dt} &= K_0 \cdot i_{Lr} - K_1 \cdot \frac{u_{Cr}}{0.01} \\ C_f \frac{du_{Cf}}{dt} + \frac{u_{Cf}}{R_l} &= i_{Lf} \end{aligned} \tag{2}$$

where

$$K_0 = \begin{cases} 0, & when\ applying\ a\ control\ signal\ to\ the\ transistor\ or\ u_{Cr} \leq 0 \\ 1, & for\ all\ other\ cases \end{cases}$$

$$K_1 = \begin{cases} 1, & for\ u_{Cr} < 0 \\ 0, & for\ all\ other\ cases \end{cases}$$

Based on the above equations, the mathematical model, which is visualized in Figure 3, is implemented in the Simulink/MATLAB environment.

The model from Figure 3 is also used for validation because on the basis of the system (2) in the next point, an author's code is compiled, which will be used to implement several optimization procedures (based on the `fminbnd` and `fgoalattain` commands). Via these optimization procedures, optimal values (according to selected criteria) of the circuit elements of the device will be found.

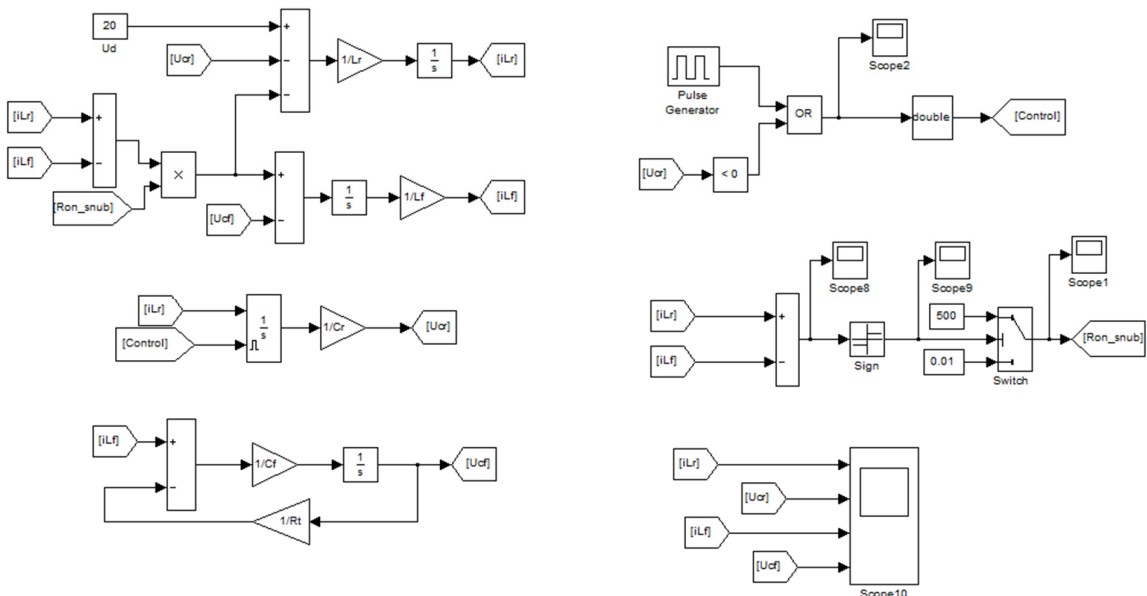

**Figure 3.** Mathematical model of the power circuit.

### 3. Initial Selection of Schematic Elements

The methodology proposed in [1,3] was used to design the Buck ZVS Quasi-Resonant DC-DC converter.

The algorithm resulting from this methodology is as follows:

1. Initially, the DC transfer function of the DC-DC converter—$M_{vdc}$ is set;
2. Determination of the quality factor of the resonant circuit—$Q = M_{vdc}$;
3. Selection of the control frequency of the converter—$f$;
4. Determination of the nominal value of the load resistance $R_{load}$;
5. Determination of the resonance frequency of the successive resonance circle—$f_0$;

$$f_0 = \frac{3f_s(\pi + 1)}{4\pi(1 - M_{\mathbf{vdc}})}$$

6. Calculation of the Duty cycle of the Quasi-Resonant DC-DC converter—$D$;

$$D = 1 - \frac{f_s(3\pi + 2)}{4\pi f_0}$$

7. Finding the value of resonant inductance—$L_r$;

$$L_r = \frac{R_l}{2\pi Q f_0}$$

8. Finding the value of the resonant capacitor—$C_r$;

$$C_r = \frac{Q}{2\pi R_l f_0}$$

A specific computational example with the following input parameters is taken to illustrate the optimization procedures:

Input voltage $U_i$ = 20 V, Output voltage $U_0$ = 10 V; $f$ = 1 MHz; Nominal value of the load resistance $R_{load}$ = 10 Ω.

After calculating the device parameters according to the presented procedure, the following values were obtained:

$D$ = 54.0242%, the value of resonant inductance $L_r$ = 1.6097 µH, and the value of the resonant capacitor $C_r$ = 4.0242 nF.

Unfortunately, according to this design procedure [1], no algorithm was proposed to calculate the filter elements. To calculate these elements, we use methodology [29,30] for calculating the filter elements of a standard Buck DC-DC converter.

The algorithm resulting from this methodology is supplemented with the following:

1.  Nominal value of the output voltage $u_o = u_i M_{\mathbf{vdc}}$;
2.  Nominal value of the output current $i_o = u_o / R_{load}$;
3.  Minimum output current value $i_{o\_min} = 0.05\, i_0$;
4.  Minimum input voltage value $u_{i\_min} = 0.8\, u_i$;
5.  Maximum input voltage value $u_{i\_max} = 1.15\, u_i$;
6.  Maximum value of the load resistance $R_{load\_max} = u_0 / i_{o\_min}$;
7.  Minimum value of the transfer function $M_{\mathbf{vdc}\_min} = u_o / u_{i\_max}$;
8.  Maximum value of the transfer function $M_{\mathbf{vdc}\_max} = u_o / u_{i\_min}$;
9.  Minimum duty cycle value $D_{\min} = M_{\mathbf{vdc}\_min} / 0.85$;
10. Maximum duty cycle value $D_{\max} = M_{\mathbf{vdc}\_max} / 0.85$;
11. Minimum value of filter inductance $L_{f\min} = R_{l\_max}\,(1 - D_{\min})/(2 f_s)$;
12. Current ripples through the filter inductance $\Delta_{il} = u_0 (1 - D_{\min})/(L_{f\min} f_s)$;
13. Output voltage ripple $u_r = 0.01\, u_o$;
14. Active resistance of the filter capacitor $R_{Cf} = u_0 / \Delta_{ilf}$;
15. Minimum value of the filter capacitor $C_{\min} = D_{\max}\,(1 - D_{\min})/(2 R_C f_s)$.

After calculating the parameters according to this procedure, we obtain the following:

The minimum value of the filter capacitor *Cf_min* = 367.65 nF and the minimum value of filter inductance *Lf_min* = 48.849 μH.

Combining these two design procedures, we can choose the following converter design parameters:

$$L_r = 1.6 \text{ μH}, \ C_r = 4 \text{ nF}, \ L_f = 0.5 \text{ μH}, \ \text{and} \ C_f = 37 \text{ μF}.$$

With the values thus obtained, the Buck ZVS Quasi-Resonant DC-DC Converter was simulated, and the results are shown in Figure 4 (from top to bottom as follows: output voltage $U_0 = U_f$, the voltage on the transistor, and resonant capacitor $u_{cr}$—for the entire duration of the process and in expanded view to demonstrate operation in ZVS mode). Analysis of the shape of the output voltage and the voltage on the resonant capacitor shows that during the transient process, larger values of both quantities are observed compared to the established mode (and accordingly set during the design).

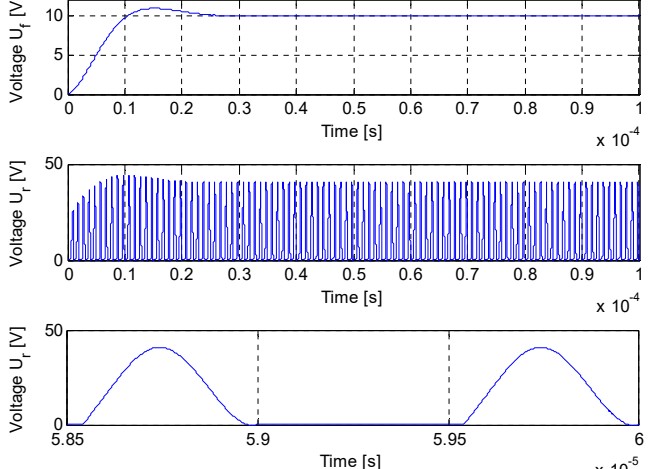

**Figure 4.** Timing diagrams from the simulation study of the Buck ZVS Quasi-Resonant DC-DC converter, from top to bottom, as follows: output voltage $U_f$, voltage across the transistor, and the resonant capacitor $u_{cr}$—for the entire duration of the process and in expanded form.

Since the above two design procedures are derived under non-matching assumptions and concern essentially different types of DC-DC converters—classical and quasi-resonant—it can be considered that the obtained results are approximate. For this, in the following points, the authors will use the above values as initial approximations, from which numerical optimization procedures will be launched aimed at better tuning the values of these parameters.

## 4. Formulation of a Two-Criteria Optimization Problem with Constraints

We will use a numerical optimization procedure to obtain the best result in a certain sense under given constraints. More precisely, we will strive to realize certain dynamics, which must be evaluated with two criteria, $J_1(x)$ and $J_2(x)$, which are as follows: root mean square error between the reference and the actually realized trajectory of the output voltage and a maximum value of the voltage across the resonant capacitor and the transistor. The constraints will be equalities type—the equations of the model (1–2) and a constraint preserving the resonance condition. There will also be restrictions on the type of inequalities—setting limits for the change in the circuit elements and a limitation ensuring a mode of continuous current through the filter inductance (a necessary condition for the correct operation of the circuit). The two criteria will depend on the selection of circuit elements $L_r$, $C_r$, $L_f$, and $C_f$ of the converter.

To determine the first criterion $J_1(x)$, a suitable reference trajectory $u_{Cf\_ref}$ is chosen, whose analytical expression is [31] as follows:

$$u_{Cf, ref} = 10(1 - e^{-t/T}), \; for \; t \in [0, \; 1 \; \times 10^{-4}] \; and \; T = 0.3 \times 10^{-5}$$

The selected trajectory will be used to search for a suitable value of the elements $L_r$, $C_r$, $L_f$, and $C_f$ so that the difference between the reference shape $u_{Cf\_ref}$ and the shape of the simulated voltage $u_{Cf}$ is minimal, i.e., to minimize functional

$$J_1 = \int\limits_0^{t_{end}} \left( u_{C_f} - u_{C_f, ref} \right)^2 dt \underset{(C_r, C_f, L_r, L_f)}{\rightarrow} \min \tag{3}$$

As the second criterion, $J_2(x)$, the maximum value of the resonant voltage $u_{Cr}$ is chosen. It is usually required that the value of the resonant voltage does not exceed two or three times the input voltage $U_d$., i.e., we will minimize the following:

$$J_2 = u_{Cr} \underset{(C_r, C_f, L_r, L_f)}{\rightarrow} \min \tag{4}$$

Thus, a two-criteria optimization problem is obtained, where a vector of objectives is minimized.

$$J = [J_1, J_2] \underset{(C_r, C_f, L_r, L_f)}{\rightarrow} \min \tag{5}$$

This optimization problem is solved under the following constraints:

- Inequalities defining the boundaries of the schematic elements:

$$C_{r,\min} \leq C_r \leq C_{r,\max}, \quad C_{f,\min} \leq C_f \leq C_{f,\max} \quad L_{r,\min} \leq L_r \leq L_{r,\max}, \quad L_{f,\min} \leq L_f \leq L_{f,\max} \tag{6}$$

Inequality—providing continuous current mode through the filter inductance:

$$i_{Cf}(t) > 0.2, \text{for } t \in [\tau, t_{end}] \tag{7}$$

Equality—the equations of the model (1–2);

Equality—ensures the preservation of the resonant frequency and, accordingly, the operating mode set during design:

$$L_r C_r = \textbf{constant} \tag{8}$$

Since it is a two-criteria optimization problem, it could not be solved using the optimization procedure "Check Against Reference" built into the Simulink/MATLAB environment, as performed in [31]. Therefore, the solution to this task requires the use of the author's code.

Moreover, we have a vector optimization problem (where the components of $J(x)$ are dependent), and therefore the solution to this problem is a set of points called the Pareto frontier. We recall that a Pareto-optimal (non-improvable) solution is one where the improvement in one objective requires the deterioration of another. Finding the entire Pareto frontier is a complex enough task; moreover, even if it is found, it will eventually be necessary to introduce an additional constraint with the help of which a single point is separated from this frontier. This is because in order to design the converter, in the end, unique values of the circuit elements must be selected. For this, we will not seek to find the entire Pareto frontier but will reformulate this problem by reducing it to several one-criteria problems, as shown in the next points.

## 5. Reduction to a One-Criteria Optimization Problem with Constraints

Reducing a two-criteria optimization problem to a one-criteria problem with constraints is a common approach in multicriteria optimization. This process involves converting one of the criteria into a constraint while the other is used as an objective function. Here are the basic steps for this conversion [32,33]:

-   Selection of objective function and constraint: The first step is to decide which of the two criteria will be the main objective function and which will be converted into a constraint. This choice depends on the priorities and the specifics of the task.
-   Constraint formulation: The criterion that is chosen to be a constraint is formulated as such. For example, if the second criterion is "minimum time", a constraint of the type "time must not exceed a certain value" can be set.
-   Objective function optimization: once one criterion is formulated as a constraint, the optimization problem becomes a standard single-criteria optimization problem, where the objective is to maximize or minimize the remaining criterion, subject to the given constraints.
-   Solution analysis: It is important to analyze the resulting solutions to ensure that they meet the requirements and expectations. Sometimes, it may be necessary to adjust the constraints or the objective function to achieve better results.
-   Use of appropriate solving methods: depending on the nature of the objective function and the constraints, various optimization methods can be applied, such as linear programming, nonlinear programming, metaheuristic methods, and others.

This approach is particularly useful when one criterion can be clearly defined as a constraint while the other is more suitable for optimization. This allows for a more precise focus on the primary criterion while ensuring that secondary requirements are also met.

### 5.1. Reduction to a One-Criteria Optimization Problem with Weights

Reducing a multi-criteria optimization problem to a single-criteria problem by using weights is a popular method known as linear combination or weighted sum. This method allows combining several criteria into a common objective function by assigning weights to the individual criteria that reflect their relative importance. Here are the basic steps:

-   Determination of weights: For each criterion, a weight is determined, which reflects its importance compared to the other criteria. These weights are usually chosen so that their sum is equal to 1.

- Criteria normalization: Before applying the weights, it is important to normalize the criteria so that they are comparable. This can be carried out by rescaling each criterion so that they have a single scale (e.g., 0 to 1).
- Formulation of the objective function: the objective function is formulated as the sum of the products of the normalized values of the criteria and their respective weights.
- Objective function optimization: once the objective function is formulated, the problem becomes a standard optimization problem where the objective is to maximize or minimize that function.

Analysis of the results: once the optimal solution has been found, it is important to analyze the results to ensure that they are satisfactory in terms of all criteria.

This method is particularly useful when it is possible to set clear priorities between different criteria and when it is desirable to find a balanced solution that takes into account all important aspects of the problem.

For this purpose, (5) is replaced by:

$$J = w_1 J_1 + w_2 J_2 \underset{(C_r, C_f, L_r, L_f)}{\rightarrow} \min \qquad (9)$$

where $w_1$ and $w_2$ are appropriately chosen weights.

The remaining components of the task (1–8) do not change.

In order to solve this task, the author's code is developed, the basis of which is the `fmincon` command, i.e.,

```
[x,Fval]=fmincon(@Opt,x0,[],[],[],[],xlb,xub,@Con,options)
```

Initial approximations $\text{x}_0$ are calculated using the method from point 3.

The functional (9) is part of `@Opt`, and the criterion $J(x)$ is replaced by a sum of squares of the form $\int (\ )^2 dt = \sum (\ )^2$.

The model Equations (1) and (2) are involved in the functions `@Con` and `@Opt`. The resonance condition (8) is also part of the `@Opt` and @Opt functions.

The continuous current limit (7) is set in the @Con function.

Constraints (6) are set using the vectors `xlb` and `xub`

After the execution of the program, the following optimal values of the circuit elements were obtained:

$$L_r = 1.6337 \text{ μH}, L_f = 32.879 \text{ μH}, C_r = 3.9175 \text{ nF}, C_f = 82.684 \text{ nF}.$$

With the values thus obtained, the result shown in Figure 5 was simulated.

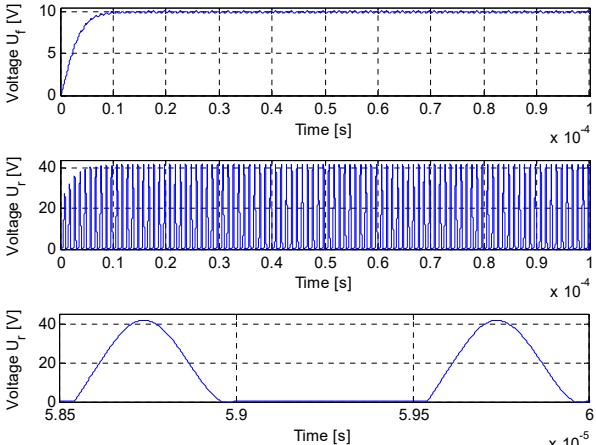

**Figure 5.** Resonance voltage $u_{Cr}$ and load voltage $u_{Cf}$ obtained for values of the circuit elements calculated with the procedure of point 5.1.

The optimization procedure from this point runs in 182 s.

*5.2. Reduction to a Single-Criteria Optimization Problem, Where the Criterion $J_2(x)$ Is Transformed into a Constraint*

After performing the optimization procedure from point 5.1, it can be seen that the maximum value of the voltage across the resonant capacitor and the transistor $u_{Cr}$ can be reduced to 41 V. For this, the two-criterion optimization problem is modified as follows:

-    The condition is added to the inequality constraints:

$$u_{Cr} < 2.1\ U_d \tag{10}$$

-    Condition (5) is replaced by:

$$J_1 \underset{(C_r,C_f,L_r,L_f)}{\to} \min \tag{11}$$

The remaining components of the task (1–8) do not change.

In order to solve this task, the author's code is developed, the basis of which is again the `fmincon` command,

```
[x,Fval]=fmincon(@Opt,x0,[],[],[],[],xlb,xub,@Con,options)
```

This code is a modification of the code used in the previous point. The differences are that condition (10) is added in the `@Con` function and that in the `@Opt` function, condition (9) is replaced by (11).

After the execution of the program, the following optimal values of the circuit elements were obtained:

$$L_r = 1.6319\ \mu H,\ L_f = 35\ \mu H,\ C_r = 3.9218\ nF,\ C_f = 100\ nF.$$

These values are very close to those of point 5.1, but this is due to the fact that condition (10) was chosen $u_{Cr} < 2.1\ U_d$. This choice is consistent with the optimum taught in point 5.1. However, if this condition were chosen $u_{Cr} < 2\ U_d$, experiments show that to achieve this, a significant deterioration of the optimal value of $J_1(x)$ is required. With the values thus obtained, the simulated results are shown in Figure 6.

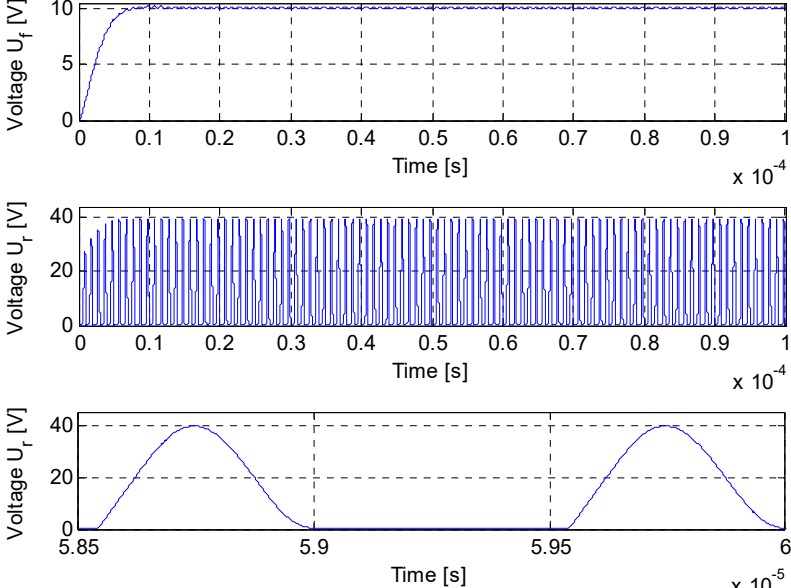

**Figure 6.** Resonance voltage $u_{Cr}$ and load voltage $u_{Cf}$ obtained for values of the circuit elements calculated with the procedure of point 5.2.

The optimization procedure from this point is executed in 30 s.

### 5.3. Reduction to a One-Criteria Optimization Problem by the Goalattain Method

Another approach to reducing the two-criteria optimization task to a single-criteria one is to use a method to achieve the goal—goalattain. This method was proposed by Gembicki [34]. It involves determining optimal design objectives, $J^* = [J_1^*, J_2^*]$, and the relative degree of underachievement or overachievement of the objectives, the latter controlled by a vector of weighting coefficients, $w = [w_1, w_2]$.

Thus, the two-criteria optimization problem is reduced to the optimization problem where

- Condition (5) is replaced by:

$$\gamma \underset{(C_r, C_f, L_r, L_f)}{\rightarrow} \min \tag{12}$$

- In this case, (10) is added to the inequality type constraints as follows:

$$J(C_1, C_2, L_3) - w\gamma \leq J* \tag{13}$$

The remaining components of the task (1–8) do not change.

This new optimization problem is solved with the command `fgoalattain`, i.e.,

```
goal = [0 0]
weight = [1 1]
[x,Fval,attainfactor] = fgo
-alattain(@Opt,x0,goal,weight,[],[],[],[],xlb,xub,@Con,options);
```

The code used to solve this task is close to the code used in point 5.1. The difference is that the `@Opt` function is reworked so that its syntax meets conditions (12) and (13). The syntax of the `@Con` function is the same as in point 5.1.

After the execution of the program, the following optimal values of the circuit elements were obtained:

$$L_r = 1.6097 \text{ μH}, L_f = 10 \text{ μH}, C_r = 4.0242 \text{ nF}, C_f = 100\text{nF}.$$

With the values thus obtained, the result shown in Figure 7 was simulated.

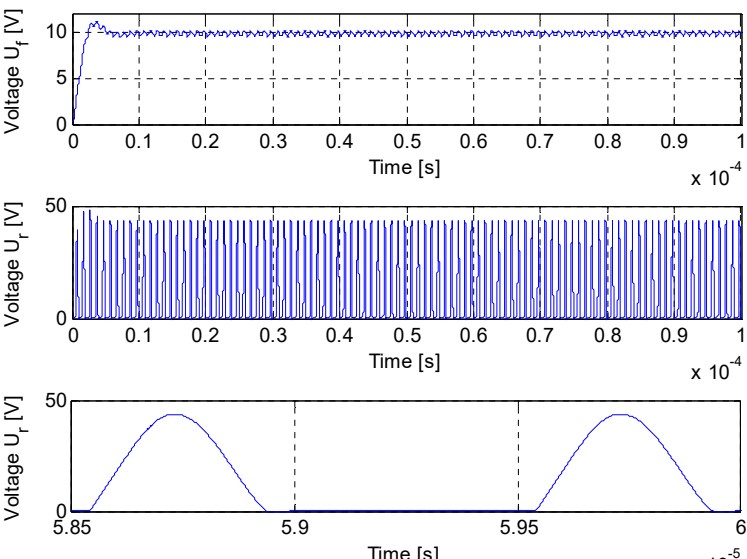

**Figure 7.** Resonant voltage $u_{Cr}$ and load voltage $u_{Cf}$ obtained for values of the circuit elements calculated with the procedure of point 5.3.

The optimization procedure from this point runs for 120 s.

We recall that Figure 3 was obtained at non-optimal values of the circuit elements, and obviously, the time constant of its transient process is of the order of $T = 0.5 \times 10^{-5}$. Since a faster transition process (with time constant) was chosen for the reference curve $u_{C_{f,ref}}$, the result of Figure 6 is obviously closer to the reference than the result of Figure 3. On the other hand, the $u_{Cf}$ readjustment obtained in Figures 3 and 6 is almost the same. While in Figures 4 and 5, there is no readjustment. In this sense, the optimizations obtained in points 5.1 and 5.2 are better than those in 5.3.

We will test the behavior of the circuit for the optimal values of the circuit elements $L_r$, $C_r$, $L_f$, and $C_f$ obtained in point 5.1 (close to those of 5.2 and better than those of 5.3) when changing the input voltage $U_d$ and the load resistance $R_{load}$.

The results of these simulations are shown in Figures 8 and 9.

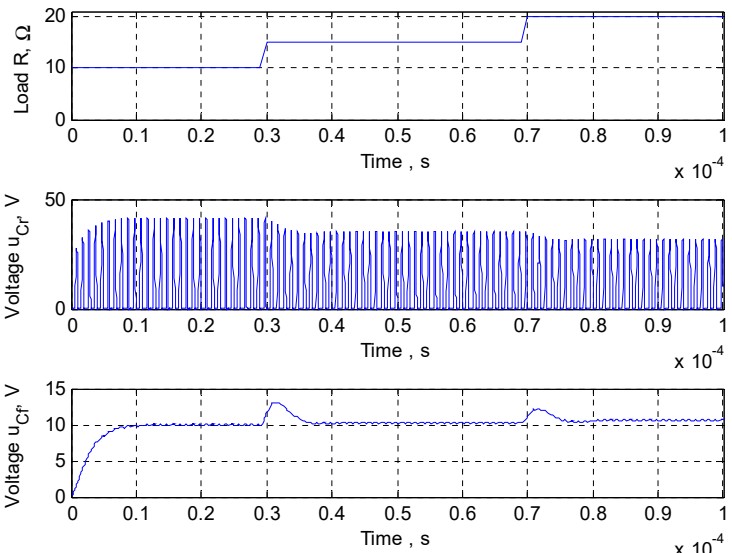

**Figure 8.** Resonance voltage and output voltage of a Buck ZVS Quasi-Resonant DC-DC converter, as the load resistance R changes.

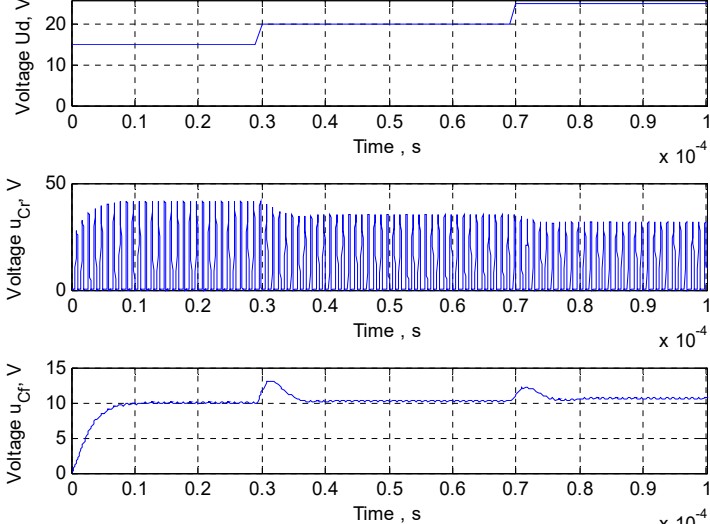

**Figure 9.** Resonance voltage and output voltage of a Buck ZVS Quasi-Resonant DC-DC converter, at changing the input voltage $U_d$.

## 6. Discussion

An analysis of the presented results shows that the best ratio between the quality of the parameters set during the optimization and the duration of the optimization procedure is obtained when using reduction, by transforming the second optimization criterion into a constraint. In this way, a remarkable ratio is achieved between the expected and obtained end result and the time for its achievement. Interesting results were obtained regarding the time duration of the optimization procedure: when using the first method, the duration is 182 s; with the second method, 30 s; and the third method, 120 s. In this sense, using a suitable optimization method reduces the calculation time by more than five times! This result is very useful in view of the optimal design of power electronic systems composed of several power electronic devices and also in the optimization of various mechatronic systems, including mechanical, electro-mechanical, and electronic parts. On the other hand, experiments by introducing disturbance on input voltage and output current show that even without using a controller, the optimally designed power electronic device has very good robustness and low sensitivity to the most common disturbing effects in DC-DC converters.

The shown example of the optimal design of the Buck ZVS Quasi-Resonant DC-DC converter gives reason to conclude that when designing power topologies where switching between individual states during operation is not only determined by external influence but also depends on the ratio of different parameters and operating mode, the combination of classical design methods and one-criteria optimization with constraints is appropriate. In this way, an optimal solution is quickly found, which meets the stated objectives to a very high degree and does not require significant software and hardware resources, as well as deep mathematical knowledge from the users.

## 7. Conclusions

Single-criteria and multi-criteria optimization problems are mathematical problems in which an optimal solution must be selected from a set of possible options, taking into account several criteria or objectives. Such tasks are found in various fields, such as engineering, economics, management, and others. Here are some of the main characteristics (advantages and disadvantages) of single-criteria and multi-criteria optimization problems:

1. Advantages of single-criteria optimization problems:
   - Easier to solve: single-criteria problems are generally easier to solve because they involve only one optimization criterion.
   - More clearly defined goals: optimizing a single criterion often yields clearly defined goals and outcomes.
   - Less computationally intensive: solving single-criteria problems usually requires less computational resources.

2. Disadvantages of single-criteria optimization problems:
   - Failure to account for real-world complexity: in real-world scenarios, there are often multiple factors to consider, and optimization of just one criterion can miss important aspects.
   - Inability to resolve conflicts: in cases where objectives are mutually contradictory, single-criteria methods may encounter difficulties in finding an optimal solution.

3. Advantages of multicriteria optimization tasks:
   - More realistic reflection of real situations: multi-criteria tasks can better reflect the complexity of the real world, where decisions often have to be balanced between different aspects.
   - Enable solutions that meet different needs: multicriteria methods allow finding solutions that meet different criteria or goals, which is important in multitasking scenarios.

4. Disadvantages of multicriteria optimization tasks:

- More complex to solve: multi-criteria problems are usually more complex to solve because of the need to balance and trade-off between different objectives.
- Requires larger computational resources: finding optimal solutions to multi-criteria problems often requires larger computational resources.

Each type of task has its place depending on the specific requirements and conditions of the problem. For example, in situations where complexity is important, and objectives are diverse, multicriteria methods are used to achieve more flexible and realistic solutions.

In this aspect, the multi-objective optimization of power electronic devices is a task that requires significant computational resources, especially when multiple possible operating modes have to be considered. As a result, various steps are often taken to simplify the optimal design task as well as to limit the size of the possible design space. This modification of the multi-criteria optimization task is performed before running a given device optimization procedure. Although this multi-criteria optimization reduction approach saves time and resources, it carries the risk of obtaining potentially suboptimal designs. In this aspect, the presented several approaches for the reduction in the optimization task are useful in order to obtain a maximally good relation between the running time and the optimality of the design.

The proposed rational optimization procedure combines the strengths of single and multi-criteria optimizations and enables the automated design of power electronic devices with complex topologies and the application of simplified computational procedures. This is important in view of the ever-higher requirements for various indicators of electronic transformers, which could not be achieved using only classical design methods. One research development is the implementation of the optimal design of the Buck ZVS Quasi-Resonant DC-DC converter with the means of machine learning.

**Author Contributions:** N.H. and B.G. were involved in the full process of producing this paper, including conceptualization, methodology, modeling, validation, visualization, and preparing the manuscript. All authors have read and agreed to the published version of the manuscript.

**Funding:** This research was funded by the Bulgarian National Scientific Fund, grant number КП-06-Н57/7/16.11.2021, and the APC was funded by КП-06-Н57/7/16.11.2021.

**Data Availability Statement:** No new data were created or analyzed in this study. Data sharing is not applicable to this article.

**Acknowledgments:** This research was carried out within the framework of the projects: "Artificial Intelligence-Based modeling, design, control, and operation of power electronic devices and systems", КП-06-Н57/7/16.11.2021, Bulgarian National Scientific Fund.

**Conflicts of Interest:** The authors declare no conflict of interest.

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
