# Peer review of "Comparison of Different Optimization Techniques for Model-Based Design of a Buck Zero Voltage Switching Quasi-Resonant Direct Current to Direct Current Converter"

_mathematics, doi:10.3390/math11244990_

Round 1

Reviewer 1 Report

Comments and Suggestions for Authors

1-Do not use abbreviations in the title of the article.

2-In this study, several optimization methods have been used to design a converter. The novelty of the study is not clear or not well presented.

3-References should be updated.

4-Explanation about past works in the introduction is not enough and you should compare with other works and your work.

5-There are many spelling mistakes in the text. Please review the text of the article.

6-Explain more about this sentence in the abstract because it is completely incomprehensible, “In this way, the optimization is performed without the need to have a complete design of the device, which is required when applying other objective functions, such as achieving minimum losses or maximum efficiency”

7-Usually, in optimization, we have a series of objective functions and optimization constraints. In this study, specify functions and constraints.

8-How did you get parameters 1 to 15 on pages 8 to 9 of the article?

9-What is your optimization method in this study? This section needs more explanations and also a flowchart

10-In sections 5-1 to 5-3, you have presented different methods in an incomprehensible way. Is this your optimization method? These sections should be presented in a more understandable and orderly manner, and additional explanations should be removed.

11-In order to validate the method, the results of the proposed model should be compared with other studies or other methods.

12-In section (6), there are explanations about the advantages and disadvantages of the model. It is better to provide these explanations with valid reasons.

Comments on the Quality of English Language

Moderate editing of English language required.

Author Response

First of all, I would like to thank you for your thorough review of our paper (mathematics-2724120) and helpful comments to improve it.

Reviewer 1

Comments and Suggestions for Authors

1-Do not use abbreviations in the title of the article.

2-In this study, several optimization methods have been used to design a converter. The novelty of the study is not clear or not well presented.

3-References should be updated.

4-Explanation about past works in the introduction is not enough and you should compare with other works and your work.

5-There are many spelling mistakes in the text. Please review the text of the article.

6-Explain more about this sentence in the abstract because it is completely incomprehensible, “In this way, the optimization is performed without the need to have a complete design of the device, which is required when applying other objective functions, such as achieving minimum losses or maximum efficiency”

7-Usually, in optimization, we have a series of objective functions and optimization constraints. In this study, specify functions and constraints.

8-How did you get parameters 1 to 15 on pages 8 to 9 of the article?

9-What is your optimization method in this study? This section needs more explanations and also a flowchart

10-In sections 5-1 to 5-3, you have presented different methods in an incomprehensible way. Is this your optimization method? These sections should be presented in a more understandable and orderly manner, and additional explanations should be removed.

11-In order to validate the method, the results of the proposed model should be compared with other studies or other methods.

12-In section (6), there are explanations about the advantages and disadvantages of the model. It is better to provide these explanations with valid reasons.

Comments on the Quality of English Language

Moderate editing of English language required.

To Reviewer 1:

            Thank you very much for your review and valuable remarks.

  1. Do not use abbreviations in the title of the article.

- Thank you very much for the remark. The abbreviation in the title has been removed.

  1. In this study, several optimization methods have been used to design a converter. The novelty of the study is not clear or not well presented.

- Thank you very much for the remark! Our goal is not to present the best approach or optimization method, because the tasks that are solved with this mathematical method are very diverse and conditioned by the different topologies and possible operating modes of the power circuits. In this sense, our goal is to propose a rational approach to reduce multi-criteria optimization to single-criteria optimization with constraints using reference curve optimization. The contribution of the paper is to demonstrate a rational optimal design approach applicable to all types of power electronic devices based on reference curve optimization and one-criteria constrained optimization. In this way, very good results are obtained from the point of view of achieving the design task, applying relatively simple procedures from a mathematical point of view, which do not require significant software and hardware resources, as well as deep mathematical knowledge from the users.

  1. References should be updated.

- Thank you very much for your comment. Necessary corrections have been made.

  1. Explanation about past works in the introduction is not enough and you should compare with other works and your work.

- Thank you very much for the suggestion! Necessary corrections have been made..

  1. There are many spelling mistakes in the text. Please review the text of the article.

- Thank you very much for reading the paper carefully! Necessary corrections have been made.

  1. Explain more about this sentence in the abstract because it is completely incomprehensible, “In this way, the optimization is performed without the need to have a complete design of the device, which is required when applying other objective functions, such as achieving minimum losses or maximum efficiency”.

- Thank you very much for the remark. I didn't express myself clearly enough. The main idea is to use as a target function in the optimization a minimum deviation from a reference curve of the output voltage, while limiting the maximum value of the voltage on the resonant capacitor and the transistor and minimum current ripples through the load. The sentence in the abstract has been edited.

  1. Usually, in optimization, we have a series of objective functions and optimization constraints. In this study, specify functions and constraints.

- Thank you very much for the remark. As a target function in the optimization, a minimum deviation from a reference curve of the output voltage was used, under limitations of the maximum value of the voltage on the resonant capacitor and of minimum current ripples through the load. When choosing a range of variation of the circuit parameters, past experience and constructive considerations related to mass, size and price indicators are taken into account.

  1. How did you get parameters 1 to 15 on pages 8 to 9 of the article?

- Thank you very much for the remark. Parameters 1 to 15 are obtained from a combination of methodology for designing a standard and quasi-resonant Buck DC-DC converter. Unfortunately, these methods are based on the analysis of established operating modes of the devices and do not take into account dynamic indicators. In this sense, the parameters determined in this way are initial approximations for conducting the various optimization procedures.

  1. What is your optimization method in this study? This section needs more explanations and also a flowchart

- Thank you very much for the remark. In the paper, the two-criteria optimization is only defined, but the solution of this type of problem is not performed. A Pareto method is usually used to solve such problems. For the transformation of two-criteria optimization into one-criteria, the following are applied: one-criteria with weights; reduction to a one-criteria optimization problem by the goalattain method.

  1. In sections 5-1 to 5-3, you have presented different methods in an incomprehensible way. Is this your optimization method? These sections should be presented in a more understandable and orderly manner, and additional explanations should be removed.

- Thank you very much for the remark. Added relevant section notes.

  1. In order to validate the method, the results of the proposed model should be compared with other studies or other methods.

- Thank you very much for the remark. The main task of the presented research is not to propose a best or productive optimization method, but rather to illustrate a rational approach to optimal design that combines classical methods based on design methodologies and simple but effective optimization techniques. Thus, our goal is to present an innovative tool for the benefit of power electronics designers and and educations in power electronics.

  1. In section (6), there are explanations about the advantages and disadvantages of the model. It is better to provide these explanations with valid reasons.

- Thank you very much for the recommendation. In this sense, a new discussion section has been added, where relevant conclusions are made.

Comments on the Quality of English Language

Moderate editing of English language required.

- А stylistic edit was made to the paper.

 Thank you very much for your remarks and comments. They were very useful for me to emphasize the main tasks and contributions of the manuscript, and also to focus the attention of the readers on the new and unique elements.

Reviewer 2 Report

Comments and Suggestions for Authors

Manuscript ID: mathematics-2724120-v1

Journal: Mathematics

Manuscript title: Comparison of Different Optimization Techniques for Model-Based Design of a Buck ZVS Quasi-Resonant DC-DC Converter

Comments to Authors:

  1. In the text, the authors should use “paper” instead “manuscript”.
  2. Some abbreviations should be identified before using in text.
  3. The proposed methods of this paper should be compared to the other methods cited in the literature in terms of simplicity, complexity, convergence, etc.
  4. What is the contribution of the paper?

Comments on the Quality of English Language

Minor editing of English language required.

Author Response

First of all, I would like to thank you for your thorough review of our paper (mathematics-2724120) and helpful comments to improve it.

Reviewer 2

Comments to the Authors

In the text, the authors should use “paper” instead “manuscript”.

Some abbreviations should be identified before using in text.

The proposed methods of this paper should be compared to the other methods cited in the literature in terms of simplicity, complexity, convergence, etc.

What is the contribution of the paper?

Comments on the Quality of English Language

Minor editing of English language required.

To Reviewer 2:

            Thank you for your review and valuable remarks.

  1. In the text, the authors should use “paper” instead “manuscript”.

- Thank you very much for your comment. Направени са необходимите корекции.

  1. Some abbreviations should be identified before using in text.

- Thank you very much for your comment. The paper has been reviewed and edited carefully, according to your note.

  1. The proposed methods of this paper should be compared to the other methods cited in the literature in terms of simplicity, complexity, convergence, etc.

- Thank you very much за вашия коментар. Our goal is not to present the best approach or optimization method, because the tasks that are solved with this mathematical method are very diverse and conditioned by the different topologies and possible operating modes of the power circuits. In this sense, our task is to propose a rational optimization approach, where multi-criteria optimization is reduced to single-criteria optimization along a reference curve, with additional constraints added..  

  1. What is the contribution of the paper?

- Thank you very much for your comment. The contribution of the paper is to demonstrate a rational optimal design approach applicable to all types of power electronic devices based on reference curve optimization and one-criteria constrained optimization. In this way, very good results are obtained from the point of view of achieving the design task, applying relatively simple procedures from a mathematical point of view, which do not require significant software and hardware resources, as well as deep mathematical knowledge from the users.

Minor editing of English language required.

             - А stylistic edit was made to the paper.

Thank you very much for your remarks and comments. They were very useful for me to emphasize the main tasks and contributions of the manuscript, and also to focus the attention of the readers on the new and unique elements.

Reviewer 3 Report

Comments and Suggestions for Authors

Comments of this reviewer on the manuscript Mathematics-2724120 are as follows:

1.     This manuscript is original, but this reviewer is not clear about the specific type of this study. Is this a review or research article?

2.     The topic considered has a mathematical background and belongs to the scope of the Mathematics journal.

3.     It is more convenient to initiate the introduction with “The present paper provides…” instead of with “The manuscript presents…”.

4.     Keywords should be in singular form and listed in alphabetical order.

5.     Since all readers of this journal are not electrical engineers, Authors are suggested to define all abbreviations when they first appear in the abstract and/or when they first appear in the remaining text.

6.     There are some typos in this manuscript. For instance: “Butsk quasi-resonant DC-DC converter” (two times in Introduction), “…of 98.7% was achieved. [6] Presents the…”, “inear feedback control”, “…Busk, Boost and Butsk-Boost.”, “posinomial functions”, “complex methods of proketing”, “con-verter with”, etc.

7.     The introduction is too long and should be shortened if possible. The paragraph of the introduction dealing with the literature review is too long and must be split into a series of normal paragraphs. Perhaps the literature review could be separated within the introduction as Sub-section 1.1.

8.     If we are talking about experiments or experimental data, then we should use the verb “validate” and the term “validation”, not the verb “verify” and the term “verification”, respectively.

9.     Conclusion and discussion should be separated into separate sections. Then, the discussion should be expanded, and the conclusions should be quantified in an appropriate manner.

10.   It seems that the optimization methods applied are presented and described in an appropriate manner.

Comments on the Quality of English Language

Minor editing of English language required

Author Response

First of all, I would like to thank you for your thorough review of our paper (mathematics-2724120) and helpful comments to improve it.

Reviewer 3

Comments to the Authors

Comments of this reviewer on the manuscript Mathematics-2724120 are as follows:

  1. This manuscript is original, but this reviewer is not clear about the specific type of this study. Is this a review or research article?
  2. The topic considered has a mathematical background and belongs to the scope of the Mathematics journal.
  3. It is more convenient to initiate the introduction with “The present paper provides…” instead of with “The manuscript presents…”.
  4. Keywords should be in singular form and listed in alphabetical order.
  5. Since all readers of this journal are not electrical engineers, Authors are suggested to define all abbreviations when they first appear in the abstract and/or when they first appear in the remaining text.
  6. There are some typos in this manuscript. For instance: “Butsk quasi-resonant DC-DC converter” (two times in Introduction), “…of 98.7% was achieved. [6] Presents the…”, “inear feedback control”, “…Busk, Boost and Butsk-Boost.”, “posinomial functions”, “complex methods of proketing”, “con-verter with”, etc.
  7. The introduction is too long and should be shortened if possible. The paragraph of the introduction dealing with the literature review is too long and must be split into a series of normal paragraphs. Perhaps the literature review could be separated within the introduction as Sub-section 1.1.
  8. If we are talking about experiments or experimental data, then we should use the verb “validate” and the term “validation”, not the verb “verify” and the term “verification”, respectively.
  9. Conclusion and discussion should be separated into separate sections. Then, the discussion should be expanded, and the conclusions should be quantified in an appropriate manner.
  10. It seems that the optimization methods applied are presented and described in an appropriate manner.

Comments on the Quality of English Language

Minor editing of English language required.

To Reviewer 3:

            Thank you for your review and valuable remarks.

  1. This manuscript is original, but this reviewer is not clear about the specific type of this study. Is this a review or research article?

- Thank you very much for the rating. The article is research, but at the beginning a detailed overview of the topic related to the optimal design of power electronic devices is presented. The aim of the present work is to present a rational procedure for optimal design that combines classical design methods with modern achievements of modeling, computational mathematics and information and communication technologies. The basic idea is that the design methodology is based on the implementation of standard tools and application programs in mathematical software widely used by engineers, such as Matlab. This allows specialists with not very deep knowledge of mathematics and programming to apply state-of-the-art tools for designing and prototyping power electronic devices and systems. On the other hand, the main idea of model-based optimization is to achieve the maximum possible from the power circuit in its design and to obtain a device with the best possible and guaranteed performance.

  1. The topic considered has a mathematical background and belongs to the scope of the Mathematics journal.

Thank you very much for your comment.

  1. It is more convenient to initiate the introduction with “The present paper provides…” instead of with “The manuscript presents…”.

- Thank you very much for the offer. Accepted unconditionally.

  1. Keywords should be in singular form and listed in alphabetical order.

- Thank you very much for the offer. Accepted unconditionally.

  1. Since all readers of this journal are not electrical engineers, Authors are suggested to define all abbreviations when they first appear in the abstract and/or when they first appear in the remaining text.

- Thank you very much for the offer. Accepted unconditionally.

  1. There are some typos in this manuscript. For instance: “Butsk quasi-resonant DC-DC converter” (two times in Introduction), “…of 98.7% was achieved. [6] Presents the…”, “inear feedback control”, “…Busk, Boost and Butsk-Boost.”, “posinomial functions”, “complex methods of proketing”, “con-verter with”, etc.

- Thank you very much for reading the paper carefully! Necessary corrections have been made.

  1. The introduction is too long and should be shortened if possible. The paragraph of the introduction dealing with the literature review is too long and must be split into a series of normal paragraphs. Perhaps the literature review could be separated within the introduction as Sub-section 1.1.

- Thank you very much for the suggestion! Corrections have been made to the structure of the paper. The main objective of the in-depth review is to highlight the differences between the methods used so far and the new reference curve optimization approach proposed by the authors.

  1. If we are talking about experiments or experimental data, then we should use the verb “validate” and the term “validation”, not the verb “verify” and the term “verification”, respectively.

- Thank you very much for reading the paper carefully! Necessary corrections have been made.

  1. Conclusion and discussion should be separated into separate sections. Then, the discussion should be expanded, and the conclusions should be quantified in an appropriate manner.

- Thank you very much for the suggestion! Corrections have been made to the structure of the paper.

  1. It seems that the optimization methods applied are presented and described in an appropriate manner.

- Thank you very much for your careful and thorough reading of our work.

 Minor editing of English language required.

             - А stylistic edit was made to the paper.

Thank you very much for your remarks and comments. They were very useful for me to emphasize the main tasks and contributions of the manuscript, and also to focus the attention of the readers on the new and unique elements.

Round 2

Reviewer 1 Report

Comments and Suggestions for Authors

The article has made significant progress and the authors have responded well to my concerns.

Comments on the Quality of English Language

Minor editing of English language required.

Author Response

First of all, I would like to thank you for your thorough review of our paper (mathematics-2724120) and helpful comments to improve it.

Reviewer 1

Comments and Suggestions for Authors

The article has made significant progress and the authors have responded well to my concerns.

Comments on the Quality of English Language

Minor editing of English language required.

To Reviewer 1:

            Thank you very much for your review and valuable remarks.

  1. The article has made significant progress and the authors have responded well to my concerns.

- Thank you very much for the remark. Your comments and remarks were very helpful for us to improve the quality of the paper!

Minor editing of English language required.

             - А stylistic edit was made to the paper.

Thank you very much for your remarks and comments. They were very useful for me to emphasize the main tasks and contributions of the manuscript, and also to focus the attention of the readers on the new and unique elements.

Reviewer 2 Report

Comments and Suggestions for Authors

- Some previous comments should be inserted in the revised paper.

Author Response

First of all, I would like to thank you for your thorough review of our paper (mathematics-2724120) and helpful comments to improve it.

Reviewer 2

Comments to the Authors

- Some previous comments should be inserted in the revised paper.

To Reviewer 2:

            Thank you for your review and valuable remarks.

  1. Some previous comments should be inserted in the revised paper.

- Thank you very much for your comment. An addition was made to the discussion section and the whole paper was re-reviewed. Unfortunately, your remarks and comments are too general and this makes it very difficult for us to revise the paper.

Thank you very much for your remarks and comments. They were very useful for us to emphasize the main tasks and contributions of the manuscript, and also to focus the attention of the readers on the new and unique elements.